# Cyclic [Cu-biRadical]_2_ Secondary Building Unit in 2p-3d and 2p-3d-4f Complexes: Crystal Structure and Magnetic Properties

**DOI:** 10.3390/molecules28062514

**Published:** 2023-03-09

**Authors:** Xiao-Tong Wang, Xiao-Hui Huang, Hong-Wei Song, Yue Ma, Li-Cun Li, Jean-Pascal Sutter

**Affiliations:** 1Key Laboratory of Advanced Energy Materials Chemistry, Department of Chemistry, College of Chemistry, Nankai University, Tianjin 300071, China; 2Laboratoire de Chimie de Coordination du CNRS (LCC-CNRS), Université de Toulouse, Centre National de la Recherche Scientifique (CNRS), 31077 Toulouse, France

**Keywords:** nitronyl nitroxide biradical, heterospin, crystal structure, magnetic properties

## Abstract

Employing the new nitronyl nitroxide biradical ligand biNIT-3Py-5-Ph (2-(5-phenyl-3-pyridyl)-bis(4,4,5,5-tetramethylimidazoline-1-oxyl-3-oxide)), a 16-spin Cu-radical complex, [Cu_8_(biNIT-3Py-5-Ph)_4_(hfac)_16_] **1**, and three 2p-3d-4f chain complexes, {[Ln(hfac)_3_][Cu(hfac)_2_]_2_(biNIT-3Py-5-Ph)_2_}_n_ (Ln^Ⅲ^= Gd **2**, Tb **3**, Dy **4**; hfac = hexafluoroacetylacetonate), have been prepared and characterized. X-ray crystallographic analysis revealed in all derivatives a common cyclic [Cu-biNIT]_2_ secondary building unit in which two bi-NIT-3Py-5-Ph biradical ligands and two Cu^II^ ions are associated via the pyridine N atoms and NO units. For complex **1**, two such units assemble with four additional Cu^II^ ions to form a discrete complex involving 16 *S* = 1/2 spin centers. For complexes **2**–**4**, the [Cu-biNIT]_2_ units are linked by Ln^III^ ions via NO groups in a 1D coordination polymer. Magnetic studies show that the coordination of the aminoxyl groups with Cu or Ln ions results in behaviors combining ferromagnetic and antiferromagnetic interactions. No slow magnetic relaxation behavior was observed for Tb and Dy derivatives.

## 1. Introduction

A particularly promising strategy for designing molecular-exchange-coupled magnetic materials is to combine organic radicals and paramagnetic metal ions [1,2,3]. Utilizing this strategy, some appealing results have been achieved, among which are high-*T*_C_ molecular-based magnets [4,5], spin-transition-like complexes [6,7,8,9,10] and molecular nanomagnets, including single-molecule magnets (SMMs) and single-chain magnets (SCMs) [11,12,13]. Typical paramagnetic ligands are based on stable radicals such as semiquinone derivatives [14,15,16], nitronyl nitroxides [17,18,19], TCNE/TCNQ^−1^ [20,21,22], verdazyl radicals [23,24] and triazyls [25,26]. These radical derivatives can coordinate or even bridge paramagnetic metal ions to produce novel structures with appealing magnetic behavior, which makes them central to developments in molecular magnetism. For example, conjugated semiquinones are widely investigated in extended magnetic systems, i.e., magnets and single-molecule magnets (SMMs), due to their direct exchange mechanism based on the delocalized character of π-electrons. Slageren’s group recently reported tetraoxolene radical-bridged dinuclear Dy/Tb complexes with improved SMM behaviors due to the strong Ln–radical magnetic coupling [27]. Zheng’s group has reported the first family of *p*-semiquinone-radical-bridged lanthanide complexes, and the Dy^III^ analog exhibits a two-step slow relaxation of magnetization behavior [28]. The electron acceptor TCNQ (TCNQ = 7,7,8,8-tetracyanoquinodimethane) has been widely used as a building block by forming a stable radical in the process of designing coordination polymers and charge-transfer complexes with magnetic and electrical conducting properties [29,30,31]. Dunbar’s group reported TCNQ-based conductive SMMs [Dy(TPMA)(µ-TCNQ)(µ-OH)](TCNQ)_2_·CH_3_CN, which are the first TCNQ–rare-earth bifunctional molecular materials with high electrical conductivity [20]. Verdazyl radicals also are employed to build molecular magnetic materials. Train et al. have obtained a six-spin cluster [(vdpy-CH_2_O)_2_Co_2_Dy_2_ac_8_] (Hac = HO_2_CCH_3_) involving verdazyl radicals that exhibit SMM behavior [32]. It should be noted that, among all of these organic radicals, the most well-documented family of metal–radical complexes involves nitronyl nitroxide-derived ligands (NIT-R) because of the high stability and facile chemical modification of this radical. Metal complexes of 3d or 4f metal ions have been studied extensively and shown to allow a remarkable diversity of molecular architectures and appealing magnetic properties [18,33,34,35]. The first SCM [Co(hfac)_2_(NITPhOMe)] is constructed by using a nitronyl nitroxide radical [11]. Notably, a very large proportion of the reported nitronyl nitroxide metal complexes involve monoradical ligands [35,36,37,38,39,40,41,42], whereas diradicals have been much less considered. However, a poly-NIT ligand possesses more coordination sites that can lead to unique assemblage topologies with metal ions and provide opportunities for the construction of novel magnetic systems. Gatteschi et al. reported a mononuclear Dy-nitronyl nitroxide biradical complex exhibiting SMM behavior [43]. A series of high-nuclear Ln-biradical SMMs were obtained by using a nitronyl nitroxide biradical involving a flexible pyridine group [44]. Wang et al. have successfully constructed Ln-based SMMs using a nitronyl nitroxide triradical [45]. More recently, 2p-4f- and 2p-3d-4f-based SCMs involving nitronyl nitroxide biradicals have been reported. Ishida and co-workers prepared a series of rare-earth chains bridged with triplet nitroxide biradicals, and a magnetic hysteresis loop was recorded for the Tb derivative [46]. Our group has also been working on the construction of SCMs using biradicals. We have shown that a biradical, with a ground spin state S = 1, will help to strengthen the spin-flip barrier (i.e., the activation energy for magnetization relaxation, U_eff_) by counteracting second-neighbor interactions between two NITs coordinated to a Ln center that is antiferromagnetic in nature. Thus, a biradical-based nitronyl nitroxide-Cu–Dy chain with U_eff_/k_B_ = 40 K could be achieved [47].

In the present paper, a new nitronyl nitroxide biradical ligand, namely, 2-(5-phenyl-3-pyridyl)-bis(4,4,5,5-tetramethylimidazoline-1-oxyl-3-oxide(bi-NIT-3Py-5-Ph) (Figure 1), is shown to lead to a particular cyclic dimer with Cu^II^ ions that acts as a secondary building unit (SBU) in a series of 2p-3d and 2p-3d-4f complexes. The assemblage of two such SBUs with additional Cu^II^ ions gave a discrete 16-spin complex, [Cu_8_ (biNIT-3Py-5-Ph)_2_(hfac)_16_] (**1**), while in the presence of Ln^III^ ions, it led to the 1D coordination polymer {[Ln(hfac)_3_][Cu(hfac)_2_]_2_(biNIT-3Py-5-Ph)_2_} _n_ (Ln= Gd, **2**; Tb, **3**; Dy, **4**). The preparation, crystal structures and magnetic behaviors have been investigated for all complexes.

## 2. Results and Discussion

### 2.1. Spectral Properties

The IR spectras of **1**–**4** are shown in Appendix A. For **1**, the absorption bands of the coligand hfac^–^ appear at 1252 cm^−1^(ν_C-F_), 1133 cm^−1^ (ν_C-F_), 670 cm^−1^ (δ_C-F_), 1646 cm^−1^(ν_C=O_) and 799 cm^−1^(δ_C-O_). The observed absorption peaks at 1527 cm^−1^ and 1368^−1^cm are assigned to C=N and N–O stretching in the biradical ligand. The IR spectra of complexes **2–4** are similar. The peaks at about 1252 cm^−1^, 1130 cm^−1^ (ν_C-F_), 659 cm^−1^ (δ_C-F_), 1650 cm^−1^(ν_C=O_) and 798 cm^−1^(δ_C-O_) are assigned to the coligand hfac^–^, while the peaks at about 1510 cm^−1^ and 1368 cm^−1^ originate from C=N and N–O stretching in the biradical ligand [35].

### 2.2. Description of the Crystal Structures

A structural feature common to all of the complexes is the occurrence of a cyclic [Cu-biNIT]_2_ moiety that acts as a secondary building unit in the formation of complexes **1**–**4**, as depicted in Figure 2. This unit results from the association of two biNIT-3Py-5-Ph ligands linked to two Cu^II^ ions by means of one of the NO groups and the pyridine nitrogen atom. The centrosymmetric [Cu-biNIT]_2_ moiety is formed even for higher Cu^II^ stoichiometry or in the presence of other ions such as Ln^III^, suggesting a preferred assembly pattern between biNIT-3Py-5-Ph and Cu^II^.

The single-crystal X-ray diffraction analysis reveals that complex **1** possesses a centrosymmetric structure (Figure 1) and crystallizes in the triclinic space group *P*ī. The asymmetric unit is composed of four Cu(hfac)_2_ units and two biNIT-3Py-5-Ph ligands. The two biradical ligands are coordinated to Cu^II^ ions in different ways: one acts as a tridentate ligand in *μ*_3_-*κ*O:*κO*: *κN* mode, and the other behaves as a tetradentate ligand in *μ*_4_-*κ*O:*κO*: *κO*:*κN* mode. Cu1 has a square-pyramidal coordination sphere, while the other three Cu^II^ ions (Cu2, Cu3 and Cu4) adopt an elongated octahedral geometry (Figure 1 and Appendix A). Cu1 is coordinated by five oxygen atoms from two hfac coligands and one NO group. One O_hfac_ atom is located in the axial site, as evidenced by the longer Cu-O_hfac_ bond (2.119(4) Å) compared to other Cu-O bond lengths (Cu-O_rad_: 1.985(4) Å; Cu-O_hfac_: 1.912(4)–1.970(4) Å). Both Cu2 and Cu4 have a {NO_5_} coordination environment. The equatorial plane includes three O_hfac_ atoms and one N atom from pyridine (Cu-O_hfac_: 1.939(4)–1.949(4) Å for Cu2, 1.942(4)–1.962(4) Å for Cu4, Cu-N: 2.005(4) Å for Cu2, 2.013(4) Å for Cu4). The axial positions are occupied by two oxygen atoms, one from a NO unit (Cu-O_rad_: 2.592(4) Å for Cu2, Cu-O_rad_: 2.489(4) Å for Cu4) and one from an hfac group (Cu-O_hfac_: 2.217(4) Å for Cu2**,** Cu-O_hfac_:2.261(4) Å for Cu4). The Cu-O-N angles are 144.1(4)°for Cu2 and 129.2(3)°for Cu4. Cu3 is six-coordinated by four oxygen atoms from hfac groups in the equatorial plane (Cu-O_hfac_: 1.927(4)–1.936(4) Å) and by two oxygen atoms from two NO groups in the axial positions. The latter bonds are longer (Cu-O_rad_: 2.464(5) and 2.50(5) Å), indicating a Jahn–Teller effect [6,48]. The Cu-O-N angles are 141.2(4)°and 121.37(3), respectively. In the molecular complex, the shortest distance between Cu ions is 7.83(1) Å, and the separation of the uncoordinated NO group is 7.18(7) Å. The crystal packing diagram of **1** is shown in Appendix A.
Figure 1Crystal structure of **1** (fluorine and hydrogen atoms are omitted for the sake of clarity; symmetry code, a: x, y, 1+z).
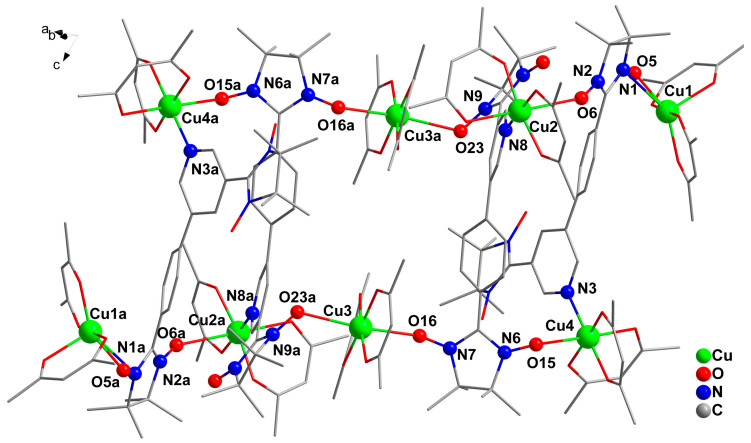

Figure 2One-dimensional structure of **2** and coordination polyhedra of Gd^Ⅲ^ ion (fluorine and hydrogen atoms are omitted for the sake of clarity).
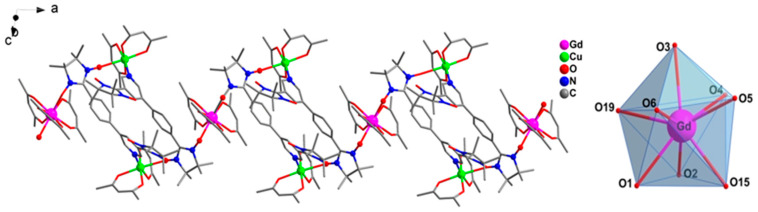


Complexes **2**–**4** are isomorphous and display a one-dimensional structure (Figure 2 and Appendix A). Therefore, only the structure of **2** is briefly described. Its asymmetric unit incorporates a Gd(hfac)_3_, two Cu(hfac)_2_ and two biNIT-3Py-5-Ph radical ligands. The [Cu-biNIT]_2_ SBU formed between biNIT-3Py-5-Ph and Cu^II^ is connected to two Gd ions via the two NIT groups already linked to Cu^II^. Likewise, each Ln center is linked to two SBUs, thus developing a zig-zag chain. Gd^III^ is eight-coordinated with two oxygen atoms from the NO groups (Gd-O_rad_: 2.367(7) Å, 2.364(7) Å) and six oxygen atoms from three chelating hfac ligands (Gd-O_hfac_: 2.358(7)–2.415(7) Å). The Gd-O-N angles are 131.9(5)° and 134.3(6)°, respectively. These distances compare well to those of the reported Ln(hfac)_3_-nitronyl nitroxide complexes [40,49]. The coordination sphere around Gd has a distorted triangular dodecahedron geometry (*D_2d_*), as revealed by SHAPE software [50,51] (Table 1). Each Cu^II^ ion is six-coordinated with one nitrogen, one oxygen from a NO unit and four oxygen atoms from hfac ligands. The equatorial Cu-O_hfac_ distances are between 1.930(6) and 1.975(7) Å, and the Cu-N distance is 2.003(8) Å. The axial positions are occupied by one oxygen atom from a NO group (Cu-O_rad_: 2.519(8) Å) and another from an hfac group (Cu-O_hfac_: 2.202(7) Å); these bonds are significantly longer because of the Jahn–Teller effect. In the chain, the Cu---Cu separation in the [Cu-biNIT]_2_ unit is 9.77(2) Å, and the Gd---Cu distances are 8.52(1) Å and 8.56(2) Å. For the biradical ligand, only one of the NIT connects with metal ions. Close proximity is found between uncoordinated NO units (3.642 Å). Earlier research has indicated that the short distance of NO---NO might lead to important magnetic interactions [52,53]; however, this depends strongly on the N-O⋯N angle (α) and the dihedral angle (β) between the [N-O⋯O-N] and [O-N-C-N-O] planes [52,53]. For **2**, the angles of α and β are 112.12° and 73.96° (Appendix A) and thus incompatible with the effective overlap of magnetic orbitals, resulting in weak exchange interactions. The crystal packing diagram of **2** is shown in Appendix A. The shortest interchain metal–metal separation is found between Cu ions with 8.00 (2) Å, and the shortest Gd---Gd and Gd---Cu distances are 16.578(9) Å and 13.84(1) Å, respectively.

### 2.3. Magnetic Properties

The macroscopic phase purity of each sample was confirmed by PXRD (Appendix A) before the magnetic studies. The temperature dependences of the molar magnetic susceptibilities (χ_M_) were recorded between 2 and 300 K (in cooling mode) with an applied magnetic field of 1 kOe; the results are plotted as χ_M_T vs. T in Figure 3, Figure 4 and Figure 5.

For **1**, the overall behavior is indicative of ferromagnetic interactions within the spin system. However, the value of χ_M_*T* obtained at 300 K (5.01 cm^3^ K mol^−1^) is clearly smaller than the expected value (6.0 cm^3^ K mol^−1^) for 16 independent *S* =1/2 spin centers (i.e., eight Cu^II^ ions plus eight NIT radicals). This value is close to the expected contribution of 4.50 cm^3^ K mol^−1^ for six free Cu^II^ ions and four monoradicals (Cu^II^: *C* = 0.375 cm^3^ K mol^−1^ and *S* = 1/2; radical: *S* = 1/2). This is indicative of some strong antiferromagnetic interactions that are indeed anticipated for the equatorial Cu-NIT coordination (vide infra) in **1**. As the temperature decreases, the value of χ_M_T progressively increases to reach 11.02 cm^3^ K mol^−1^ at 2 K, which shows that ferromagnetic interactions are also operative in the system. The *M* vs. *H* behavior recorded at 2.0 K for fields up to 50 kOe (Figure 3) shows a very fast increase at a low field and then tends to saturate. At 50 kOe, the magnetization reaches 11.0 *N*β, which is close to the 12 *N*β expected for twelve *S* =1/2 spins, thus confirming that the contributions of four *S* = 1/2 centers have been counteracted by strong antiferromagnetic interactions.

In **1**, the magnetic exchange, if any, between the Cu^II^ ion and the NO group spaced by a pyridine and a benzene ring must be very weak, and thus, this exchange pathway can be ignored. Therefore, the magnetic behavior of this complex can be attributed to the two spin sequences comprising eight *S* = 1/2 centers in the exchange interaction, i.e., [Cu1-Rad1-Cu2-Rad2-Cu3-Rad3-Cu4-Rad4] (Figure 3). To describe the behavior, three exchange interactions must be considered, namely, *J*_1_, between the Cu^II^ ions and the NO groups in the equatorial plane; *J*_2_, between the Cu^II^ ion and the axially coordinated NO group; and *J*_3_, accounting for the interaction between Cu^II^ ions and the NO group through the pyridine ring.

PHI software was employed to simultaneously analyze the χ_M_*T* vs. *T* and *M* vs. *H* behaviors [54]. The best fit to the experimental data gave *J*_1_ = −350 (1) cm^−1^, *J*_2_ = 25.0 (3) cm^−1^, *J*_3_ = 2.10 (3) cm^−1^, *g_Cu_* = 2.03 (1) and *g_rad_* = 2.0 (fixed). The value obtained for *J_1_* confirms a strong and antiferromagnetic Cu-NO_eq_ interaction, which can be attributed to the effective overlap of the magnetic orbital (d_x_^2^_-y_^2^) of the Cu^II^ ion and the magnetic π* orbital of the radical [1]. The obtained value (−350 cm^−1^) is comparable to those reported for the equatorial coordination of NIT to Cu^II^ [55,56,57,58]. The positive value for *J_2_* confirms the anticipated ferromagnetic Cu-NO_axial_ interactions resulting from the orthogonality of the magnetic orbitals (d_x_^2^_-y_^2^) of Cu^II^ and π* of the radical [59]. The found strength for *J_2_* is consistent with the Cu-NO exchange interactions reported for similar Cu-NIT complexes [6,60,61]. The small *J_3_* accounts for the ferromagnetic interaction due to spin polarization between Cu^II^ and NIT through the pyridine ring [62,63].

For complexes **2**–**4**, the χ_M_*T* products obtained at 300 K are, respectively, 10.27 cm^3^ K mol^−1^, 15.30 cm^3^ K mol^−1^ and 16.87 cm^3^ K mol^−1^, in good agreement with the theoretical values (10.13 cm^3^ K mol^−1^ for **2**, 14.07 cm^3^ K mol^−1^ for **3** and 16.42 cm^3^ K mol^−1^ for **4**) for one Ln^III^ ion (Gd^III^: *^8^S_7/2_*, *g* = 2, *C* = 7.88 cm^3^ K mol^−1^; Tb^III^: *^7^F_6_*, *g* = 3/2, *C* = 11.82 cm^3^ K mol^−1^; Dy^III^: *^6^H_15/2_*, *g* = 4/3, *C* = 14.17 cm^3^ K mol^−1^), two Cu^II^ ions (*S* = 1/2, *g* = 2, *C* = 0.375 cm^3^ K mol^−1^) and four radicals (*S*= 1/2, g=2, *C* = 0.375 cm^3^ K mol^−1^) in the absence of magnetic exchange. For **2**, the χ_M_*T* value increases gradually as the temperature is lowered from 300 K to 2 K, reaching a maximum of 15.89 cm^3^ K mol^−1^ (Figure 4). Such behavior is indicative of dominant ferromagnetic interactions in the system.

As mentioned above, the magnetic exchange interaction between Cu^II^ and the NIT radical through the phenyl and pyridine rings can be ignored. Thus, the magnetic properties of **2** mainly result from the exchange-coupled [Rad1-Cu1-Rad2-Gd-Rad3-Cu2-Rad4] spin sequence. The pertinent paths for magnetic interactions are shown in Figure 4, where *J*_1_ accounts for the Gd-NO exchange, *J*_2_ accounts for the Cu-NO_axial_ interaction, *J*_3_ accounts for the NO---Cu coupling via the pyridine ring, and *J_4_* accounts for the next-neighbor interaction between the two NO units coordinated to the Gd^III^ ion.

The analysis of the χ_M_*T* vs. *T* behavior with PHI gave *J_1_* = 1.41(3) cm^−1^, *J_2_* = 20.3 (3) cm^−1^, *J_3_* = 1.0 (1) cm^−1^, *J_4_* = −6.4(1) cm^−1^ and *g_cu_* = 2.03(1), with *g_rad_* = *g_Gd_* = 2.0 (fixed). The positive value for *J_1_* indicates that the Gd-NO interaction is mainly ferromagnetic, which is attributed to a charge transfer from the π* orbital of the radical to the empty 5d/6s orbitals of Gd, resulting in the spins of 4f and 5d (or 6s) orbitals being arranged in parallel [64,65,66]. The found weak ferromagnetic is in agreement with results for related Gd–NIT complexes [65,67,68]. The value obtained for *J_2_* confirms the ferromagnetic Cu-NO_axial_ interaction already discussed for **1**, as does the positive *J_3_* between the Cu^II^ ion and the NO group. It is satisfying to find that **1** and **2** have very similar values for these exchange interactions. Finally, the antiferromagnetic interaction obtained for *J_4_* is consistent with values reported in the literature [69,70].

The *M* vs. *H* behavior was recorded at 2.0 K in a field range of 0–50 KOe (Appendix A). For **2**, it is characterized by a very fast increase in magnetization for a low field and a smoother increase above 10 kOe to reach 12.4 Nβ at 50 kOe, close to the expected saturation value of 13 Nβ.

For **3**, χ_M_*T* smoothly decreases when *T* is reduced to 10 K and then rises rapidly to reach 18.1 cm^3^ K mol^−1^ at 2 K (Figure 5). Such behavior is again indicative of ferromagnetic contributions at a low temperature, in addition to the crystal field effect applying to the Tb^III^ ion. For **4**, the value of χ_M_*T* decreases steadily from 300 to 2 K, reaching 11.85 cm^3^ K mol^−1^ (Figure 5). For this derivative, the anticipated ferromagnetic Dy-ON contributions are not revealed. This overall behavior is found when the decrease due to the crystal field contribution is larger than the component of χ_M_*T* induced by weak ferromagnetic interactions [71]. For **3** and **4**, M vs. *H* behaviors reach values of 9.5 Nβ for **3** at 50 kOe and 10.8 Nβ for **4** at 70 kOe (Appendix A). AC magnetic susceptibility measurements for **3** and **4** performed without and with an applied static field showed no out-of-phase (*χ*″) signals, thus excluding the slow relaxation of the magnetization for these derivatives (Appendix A).

## 3. Materials and Methods

### 3.1. Materials and Physical Measurements

All solvents and chemicals used in the synthesis were of analytical grade. The bi-NIT-3Py-5-Ph biradical ligand was synthesized following literature methods [72,73], and the specific synthesis process of the bi-NIT-3Py-5-Ph radical ligand is shown in Scheme S1. Elemental analysis was performed on a PerkinElmer 240 elemental analyzer. FT-IR data were obtained by using a Bruker-Vector 22 Spectrophotometer. Magnetic measurements were performed on a SQUID MPMS XL-5 and VSM magnetometer, in which samples containing Dy and Tb ions were mixed with grease to avoid orientation effects. The data were corrected for the diamagnetic contributions of the sample holder and for all of the constituent atoms using Pascal’s table [74].

### 3.2. Synthesis of the Complexes

#### 3.2.1. [Cu_8_ (biNIT-3Py-5-Ph)_4_(hfac)_16_] (**1**)

Cu(hfac)_2_ (9.6 mg, 0.02 mmol) was dissolved in hot *n*-hexane (16 mL). Then, a CH_2_Cl_2_ solution (5.0 mL) of bi-NIT-3Py-5-Ph (3.1 mg, 0.01 mmol) was slowly added. The stirred solution was refluxed for 20 min, then cooled to room temperature and filtered. Slow evaporation of the filtrate at room temperature yielded green block-like crystals of **1**, which were isolated after 3 days (m = 7.8 mg, Yield: 55%). Elem. Anal. found (calcd) for C_180_H_140_Cu_8_F_96_N_20_O_48_ **1** (%): C 38.04, H 2.48, N 4.93; Found: C 38.21, H 2.28, N 4.39. IR (cm^−1^): 1646 (s), 1527 (m), 1458 (s), 1368 (m), 1252 (s), 1199 (s), 1133 (s), 799 (s), 670 (s), 590 (s), 528 (m).

#### 3.2.2. {[Gd(hfac)_3_][Cu(hfac)_2_]_2_(biNIT-3Py-5-Ph)_2_}_n_ (**2**)

A solution of Gd(hfac)_3_·2H_2_O (8.2 mg, 0.01 mmol) and Cu(hfac)_2_ (9.6 mg, 0.02 mmol) in heptane (18 mL) was heated to reflux for 3 h. Then, a CHCl_3_ solution (10 mL) of bi-NIT-3Py-5-Ph (6.2 mg, 0.02 mmol) was slowly added. The resulting solution was further refluxed for 15 min, then cooled to room temperature and filtered. The filtrate evaporated slowly at room temperature over a period of 4 days to give gray-purple strip crystals of the chain complex. For **2**: m = 14.7 mg, Yield = 55%. Elem. Anal. found (calcd) (%) for C_85_H_69_Cu_2_F_42_GdN_10_O_22_: C 38.31, H 2.61, N 5.26; Found: C 38.04, H 2.10, N 5.48. IR (cm^−1^): 1650 (s), 1510 (m), 1368 (m), 1253 (s), 1200 (s), 1138 (s), 912 (m), 869 (m), 798 (s), 659 (s), 585 (s), 545 (m).

#### 3.2.3. {[Tb(hfac)_3_][Cu(hfac)_2_]_2_(biNIT-3Py-5-Ph)_2_}_n_ (**3**)

Samples of Tb(hfac)_3_·2H_2_O (8.2 mg, 0.01 mmol) and Cu(hfac)_2_ (9.6 mg, 0.02 mmol) were dissolved in boiling n-heptane (20 mL) and heated to reflux for 4 h. Then, a CHCl_3_ solution (10 mL) of bi-NIT-3Py-5-Ph (6.2 mg, 0.02 mmol) was added. The obtained solution was refluxed for 12 min. The solution was filtered, allowed to cool down and evaporated at room temperature for 4 days. Gray-purple strip crystals of the chain complexes were obtained. For **3**: m = 15.47 mg, Yield = 58%. Elem. Anal. found (calcd) (%) for C_85_H_69_Cu_2_F_42_TbN_10_O_22_: C 38.29, H 2.61, N 5.25; Found: C 38.13, H 2.36, N 5.42. IR (cm^−1^): 1650 (s), 1508 (m), 1368 (m), 1250 (s), 1200 (s), 1137 (s), 913 (m), 870 (m), 798 (s), 659 (s), 585 (s), 545 (m).

#### 3.2.4. {[Dy(hfac)_3_][Cu(hfac)_2_]_2_(biNIT-3Py-5-Ph)_2_}_n_ (**4**)

Dy(hfac)_3_·2H_2_O (8.2 mg, 0.01 mmol) and Cu(hfac)_2_ (9.6 mg, 0.02 mmol) dissolved in hot n-heptane(20 mL) for refluxing were added to a CHCl_3_ solution (12 mL) of bi-NIT-3Py-5-Ph (6.2 mg, 0.02 mmol) with further reflux for 20 min. The solution was filtered and allowed to stay at room temperature for 3 days. Well-shaped gray-purple strip crystals suitable for X-ray structure determination were obtained. For **4**: m = 14.42 mg, Yield = 54%. Elem. Anal. found (calcd) (%) for C_85_H_69_Cu_2_F_42_DyN_10_O_22_: C 38.24, H 2.60, N 5.24; Found: C 38.05, H 2.30, N 5.56. IR (cm^−1^): 1650 (s), 1509 (m), 1370 (m), 1250 (s), 1200 (s), 1130 (s), 906 (m), 870 (m), 798 (s), 659 (s), 582 (s), 528 (m).

### 3.3. X-Ray Structure Determination

Single-crystal X-ray data of complexes **1**–**4** were collected at 113 K on a Rigaku Saturn CCD diffractometer with graphite monochromated Mo-Kα radiation (λ = 0.71073 Å). The single-crystal structure was solved by SHELXL-2014 [75] and SHELXS 2014 [76] programs. All non-hydrogen atoms were refined anisotropically, and the H atoms of organic molecules were positioned geometrically. SIMU, DELU, ISOR and other commands were used to correct some disordered C and F atoms. Crystallographic data for complexes **1**–**4** are shown in Table 2. Key bond lengths and angles are listed in Table 3, Table 4 and Appendix A, respectively.

## 4. Conclusions

The reported diradical ligand was found to interact with Cu^II^ ions to form a specific cyclic [CubiNIT)]_2_ dimer, and this unit then acts as an SBU in the formation of polynuclear complexes. A 16-spin Cu-NIT complex and a series of 2p-Cu-Ln 1D complexes were derived from this particular association pattern induced by the biNIT-3Py-5-Ph ligand. In the 2p-3d complex, the biNIT-3Py-5-Ph diradical ligands behave as three- or four-dentate ligands to bind Cu^II^ ions, leading to an octanuclear structure. In the 2p-3d-4f complexes, the cyclic [Cu-radical]_2_ dimers connect Ln ions via NO groups to develop a 1D coordination polymer. Magnetic studies have shown that there exist strong antiferromagnetic and ferromagnetic interactions stemming from the Cu-NO_eq_ and Cu-NO_axial_ moieties, respectively, in complex **1**. For **2**–**3**, the magnetic behaviors are governed by ferromagnetic NO-Ln/Cu interactions. This work demonstrates that nitronyl nitroxide diradicals containing functional groups are appealing ligands for constructing novel radical–metal complexes with interesting spin topologies.

## Data Availability

Data is contained within the article or Appendix A.

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
