# Peer review of "Cyclic [Cu-biRadical]2 Secondary Building Unit in 2p-3d and 2p-3d-4f Complexes: Crystal Structure and Magnetic Properties"

_molecules, 2023, doi:10.3390/molecules28062514_

Round 1

Reviewer 1 Report

1. I did not suggest this keyword: 2p-3d; 2p-3d-4f; please use a new one.

2. Table 1 could be removed into ESI.

3. “Complex” or “compound” should be unified.

4. In the introduction section “…spin transition-like complexes…” This could be updated the related documents, such as Dalton Trans., 2017,46:15178-15180 and J. Solid State Chem. 318(2023) 123713.

5. Give the symmetric code for the atoms in Fig.1.

6. The author only list the data for IR, it should be discussed and provide the figure.

7. The authors should highlight similar work and compare them on the J values.

Author Response

Response to Reviewer 1 Comments

Point 1: I did not suggest this keyword: 2p-3d; 2p-3d-4f; please use a new one.

Authors: According to the referee’s suggestion, we have replaced the keywords 2p-3d and 2p-3d-4f with heterospin.

Point 2: Table 1 could be removed into ESI.

Authors: Since crystal structures have a central role in the reported studies, we believe it is better to keep Table 1 in the main text as it provides the required support to evaluate the accuracy of the structure solutions.

Point 3: “Complex” or “compound” should be unified as complex.

Authors: According to the referee’s suggestion, we have changed the term "compound" to "complex".

Point 4: In the introduction section “…spin transition-like complexes…” This could be updated the related documents, such as Dalton Trans., 2017,46:15178-15180 and J. Solid State Chem. 318(2023) 123713.

Authors: We have consulted the two references suggested by the reviewer but we find that the specific contents of these reports are not at all related with the “spin transition-like complexes”. The first is dealing with long-range ordering in a Cu chain and the second is dealing with Co (II) compounds. Consequently, it is obscure to us while they should be cited and we did not follow this request.

Point 5: Give the symmetric code for the atoms in Fig.1.

Authors: Thanks for the referee’s suggestion. The symmetric code of the atoms has been given in the caption of Fig.1.       

Point 6: The author only listed the data for IR, it should be discussed and provide the figure.

Authors: According to the referee’s suggestion, we have added the IR spectra to the ESI and a short discussion has been added to the main text.

Point 7: The authors should highlight similar work and compare them on the J values.

Authors: Thanks for the referee’s suggestion. In the revised manuscript, the obtained J values have been compared with the literature and the relevant literatures have been cited (see references 61-63, 65-66, 70, 72, 73, etc…).

Reviewer 2 Report

This paper considers the coordination polymers containing 3d, 4f and 2p radical centers, based on new nitronyl nitroxide biradical ligand. It reports on structures and magnetic properties and it is concise, interesting and pleasant to read.

There are some minor points listed below.

References given for the preparation of the ligand [48,49] describe compounds that are quite distant analogues of it. Probably ligand synthesis should be added to experimental section, or more relevant citation should be added.

Spin Hamiltonian parameters derived from fitting of experimental data are better to be reported along with their uncertainties.

Caption of figure 5 seems to be inconsistent.

Author Response

Response to Reviewer 2 Comments

Point 1: References given for the preparation of the ligand [48,49] describe compounds that are quite distant analogues of it. Probably ligand synthesis should be added to experimental section, or more relevant citation should be added.

Authors: The synthesis for the radical moiety follows a procedure that is mainly same whatever the remaining part of the molecule and this is given in the references 48 and 49, thus the general scheme given in ESI (Scheme S1) is sufficiently explicit.

Point 2: Spin Hamiltonian parameters derived from fitting of experimental data are better to be reported along with their uncertainties.

Authors: According to the referee’s suggestion, the fitting parameters are now given with their uncertainties in the main text.

Point 3: Caption of figure 5 seems to be inconsistent.

Authors: Sorry, that is due to our carelessness. The caption of figure 5 has been corrected.

Reviewer 3 Report

Title: Cyclic [Cu-biRadical]2 Secondary Building Unit in 2p-3d and 2p-3d-4f Complexes: Crystal Structure and Magnetic Properties

Authors: Xiao-Tong Wang, Xiao-Hui Huang, Hong-Wei Song, Yue Ma, Li-Cun Li, Jean-Pascal Sutter

In this article, a new biradical nitronyl nitroxide ligand was found to form a specific cyclic dimer with Cu II ions and this unit then acts as the SBU in the formation of polynuclear compounds. A set of two such SBUs with additional Cu II ions gives a discrete 16-spin complex, while in the presence of Ln III ions it leads to a 1D-coordination polymer. Preparation, crystal structures and magnetic behaviors were investigated for all compounds.

The paper is well organized and clearly written. I propose to publish it after correcting the technical errors:

1. Lines 216-222 are on the left and overlap some of the data in Table 4.

2. At line 270, the value g is missing.

Author Response

Response to Reviewer 3 Comments

Point 1: Lines 216-222 are on the left and overlap some of the data in Table 4.

Authors: Thanks for the referee’s suggestion. We have reformatted the table.

Point 2: At line 270, the value g is missing.

Authors: We have added the g value in relevant place.
